# Identification of responsible sequences which mutations cause maternal *H19*-ICR hypermethylation with Beckwith–Wiedemann syndrome-like overgrowth
Satoshi Hara [1] ✉, Fumikazu Matsuhisa[2], Shuji Kitajima[2], Hitomi Yatsuki[1], Musashi Kubiura-Ichimaru[1], Ken Higashimoto[1] & Hidenobu Soejima [1] ✉

Beckwith-Wiedemann syndrome (BWS) is caused by a gain of methylation (GOM) at the imprinting control region within the *Igf2-H19* domain on the maternal allele (*H19*-ICR GOM). Mutations in the binding sites of several transcription factors are involved in *H19*-ICR GOM and BWS. However, the responsible sequence(s) for *H19*-ICR GOM with BWS-like overgrowth has not been identified in mice. Here, we report that a mutation in the SOX-OCT binding site (SOBS) causes partial *H19*-ICR GOM, which does not extend beyond CTCF binding site 3 (CTS3). Moreover, simultaneously mutating both SOBS and CTS3 causes complete GOM of the entire *H19*-ICR, leading to the misexpression of the imprinted genes, and frequent BWS-like overgrowth. In addition, CTS3 is critical for CTCF/cohesin-mediated chromatin conformation. These results indicate that SOBS and CTS3 are the sequences in which mutations cause *H19*-ICR GOM leading to BWS-like overgrowth and are essential for maintaining the unmethylated state of maternal *H19*-ICR.

Genomic imprinting is an epigenetic mechanism that regulates parent-of-origin-specific expression of certain genes in mammals. The expression of imprinted genes is regulated by cis-regulatory elements known as imprinted control regions (ICRs). Most ICRs overlap differentially methylated regions (DMRs) wherein the CpG-rich region is specifically methylated paternally or maternally. DMRs are classified as either germline/primary DMRs or somatic/secondary DMRs. DNA methylation of germline DMRs is established during gametogenesis and maintained after fertilization, whereas the unmethylated state of the opposite alleles is maintained throughout embryonic development. On the other hand, methylation of somatic/secondary DMRs is established after implantation[1]. The aberrant hypermethylation of unmethylated alleles in DMRs leads to the disruption of allele-specific expression of imprinted genes. In some cases, it also leads to defects in embryonic development and congenital diseases known as imprinting disorders[2].

The *Igf2-H19* domain is located on human chromosome 11 and mouse chromosome 7. *IGF2* encodes a paternally expressed insulin-like growth factor II. *H19* encodes a maternally expressed putative tumor suppressor noncoding RNA including microRNA[3–5]. The allele-specific expression of these imprinted genes is cis-regulated by the upstream region of *H19* (*H19*-ICR), which is highly methylated and unmethylated on the paternal and maternal alleles, respectively[6]. Loss of methylation in paternal *H19*-ICR leads to repression of *IGF2* and biallelic expression of *H19*, resulting in Silver-Russell syndrome (SRS; OMIM 180860), which is characterized by a small for gestational age, relative macrocephaly at birth, severe feeding difficulties, and low body mass index[7]. In contrast, gain of methylation at the maternal *H19*-ICR (*H19*-ICR GOM) causes biallelic expression of *IGF2* and repression of *H19*. This leads to Beckwith-Wiedemann syndrome (BWS; OMIM 130650), an overgrowth disorder characterized by macroglossia, macrosomia, abdominal wall defects, and childhood cancers such as Wilms tumor[8]. Therefore, maintaining DNA methylation in paternal *H19*-ICR and

[1]Division of Molecular Genetics and Epigenetics, Department of Biomolecular Sciences, Faculty of Medicine, Saga University, Saga, 849-8501, Japan. [2]Division of Biological Resources and Development, Analytical Research Center for Experimental Sciences, Saga University, Saga, 849-8501, Japan. ✉e-mail: shara@cc.saga-u.ac.jp; soejimah@cc.saga-u.ac.jp

protecting the unmethylated state of maternal *H19*-ICR from de novo DNA methylation are important for fetal growth and development. In addition, the somatic DMRs in the *IGF2-H19* domain, which are *IGF2*-DMR0 (corresponding to *Igf2*-DMR1 in mice), *IGF2*-DMR2, and *H19*prom (a proximal promoter region of *H19*), are also involved in *IGF2* and *H19* expression[9–11].

Two putative molecular mechanisms involved in the protection of the unmethylated state of maternal *H19*-ICR against de novo methylation have been identified. The first is the binding of CCCTC-binding factor (CTCF), a DNA methylation-sensitive chromatin insulator protein, to its unmethylated binding sites (CTSs) on the maternal allele. *H19*-ICR contains several CTSs, four in mice and seven in humans[12,13]. Binding of CTCF and cohesin complex to maternal *H19*-ICR forms allele-specific chromatin loops that insulate the 3′ enhancer downstream of *H19*[14]. Maternal transmission of mutant *H19*-ICR with disrupted four CTSs (CTS1-4) causes aberrant hypermethylation in the entire maternal *H19*-ICR during post-implantation development, resulting in BWS-like overgrowth in mice[15]. These indicate that CTCF binding at CTS1-4 is essential for the protection of the unmethylated state of maternal *H19*-ICR. The second essential step is the binding of the pluripotent factors SOX2 and OCT4 to their binding site (SOBS). In humans, point mutation in the SOBS within the maternally inherited *H19*-ICR has been observed in some patients with BWS caused by *H19*-ICR GOM[16–18]. This suggests that SOBS-bound SOX2/OCT4 protects the unmethylated state of maternal *H19*-ICR and that SOBS is the sequence responsible for BWS by *H19*-ICR GOM[19]. Unlike humans, mice with a deletion or a point mutation at the OCT-binding sites within maternal SOBS showed partial *H19*-ICR GOM, but no BWS-like phenotypes[20,21]. These

suggest that OCT4 cooperates with CTCF to protect the unmethylated state and that other responsible sequence(s) exist in the mouse *H19*-ICR (Fig. 1a).

So far, several mutant mice carrying a deletion in maternal *H19*-ICR have been reported. Maternal transmission of a mutant allele, in which a 1.8 kb region including SOBS, CTS3 and CTS4 was deleted, caused BWS-like overgrowth, but the remaining CTSs (CTS1 and CTS2) on the maternal allele remained unmethylated[22]. In contrast, mice with maternally inherited a mutant allele, in which approximately 70% of the DNA sequences including CTS2, CTS3, and SOBS in *H19*-ICR was deleted, exhibited little to no BWS-like phenotype and the remaining CTSs (CTS1 and CTS4) on the maternal allele remained unmethylated[23]. Overall, molecular mechanisms that regulate the unmethylated state of the ICR have not been clearly explained by SOBS and CTS1-4.

In this study, we aimed to identify the sequence(s) in which mutations cause *H19*-ICR GOM with BWS-like phenotypes for elucidating the mechanisms that regulate the unmethylated state of maternal *H19*-ICR. We showed that SOBS and CTS3 are essential for the unmethylated state of maternal *H19*-ICR and are responsible sequences for *H19*-ICR GOM leading to imprinting defects and BWS-like overgrowth.

## Results

### The effect of SOBS mutation in the maternal *H19*-ICR

We generated mice carrying a point mutation at SOBS using the CRISPR/Cas9 system to determine the exact extent of the hypermethylated region in the mutant *H19*-ICR. We designed an oligodeoxynucleotide (ssODN) with point mutations in both SOX- and OCT-binding sites (mSO, Fig. 1a and Supplementary Fig 1). Genotyping analysis of neonates conceived through

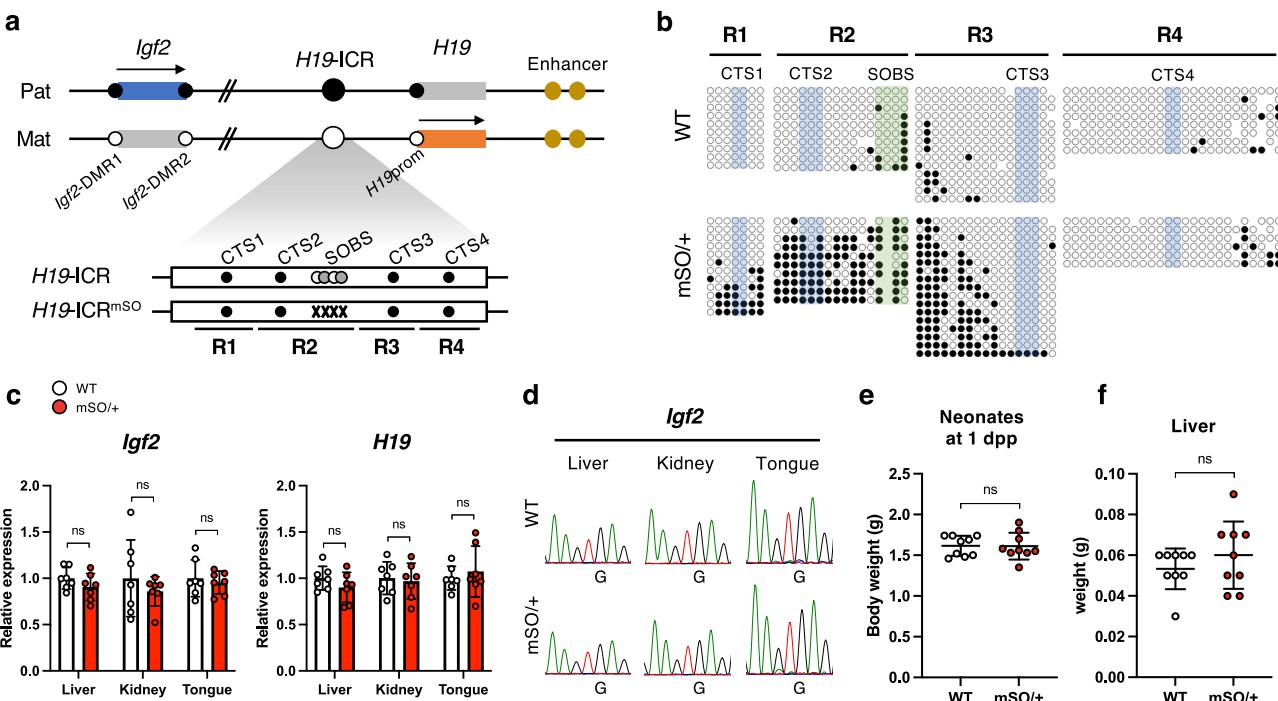

**Fig. 1 | The effect of SOBS mutation on maternal *H19*-ICR in mice. a** Schematic representation of the mouse *Igf2-H19* domain. (top) Genomic DNA is indicated by a black line, whereas blue and red boxes indicate the imprinted genes expressed in paternal and maternal alleles, respectively. Arrows indicate the direction of transcription and expressed alleles. Black and while circles indicate methylated and unmethylated regions (*Igf2*-DMR1, *Igf2*-DMR2, *H19*-ICR, and *H19*prom), respectively. (bottom) The white box shows *H19*-ICR. Black, white, and gray circles demonstrate the binding sites of CTCF (CTS1-4), SOX2, and OCT4 (SOBS), respectively. Gray Xs indicate mutated SOBS. Black lines indicate amplified regions for DNA methylation analysis (R1–4). **b** Methylation status at maternal *H19*-ICR in WT and *H19*-ICR^mSO/+ neonates. Representative results of maternal alleles are shown. Black and white circles indicate methylated and unmethylated CpGs,

respectively. CpG sites included in CTS1-4 and SOBS are highlighted in blue and green, respectively. Full results are shown in Supplementary Fig. 2. **c** Expression levels of *Igf2* and *H19* in *H19*-ICR^mSO/+ neonates. White and red bars indicate WT and *H19*-ICR^mSO/+ tissues, respectively (WT and *H19*-ICR^mSO/+; *n* = 7 and *n* = 7, respectively, from 3 litters). Error bars indicate standard deviation (SD). **d** Allelic expression of *Igf2* in *H19*-ICR^mSO/+ neonates. Representative electropherograms are shown. **e, f** The body and liver weights in WT and *H19*-ICR^mSO/+ neonates at 1 dpp (WT and *H19*-ICR^mSO/+; *n* = 9 and *n* = 9, respectively, from 3 litters). White and red dots indicate individual weights of WT and heterozygous mutants with maternally transmitted *H19*-ICR^mSO alleles, respectively. The mean ± SD are shown in a horizontal bar and error bars, respectively. ns; not significant (unpaired two-tailed *t*-test).

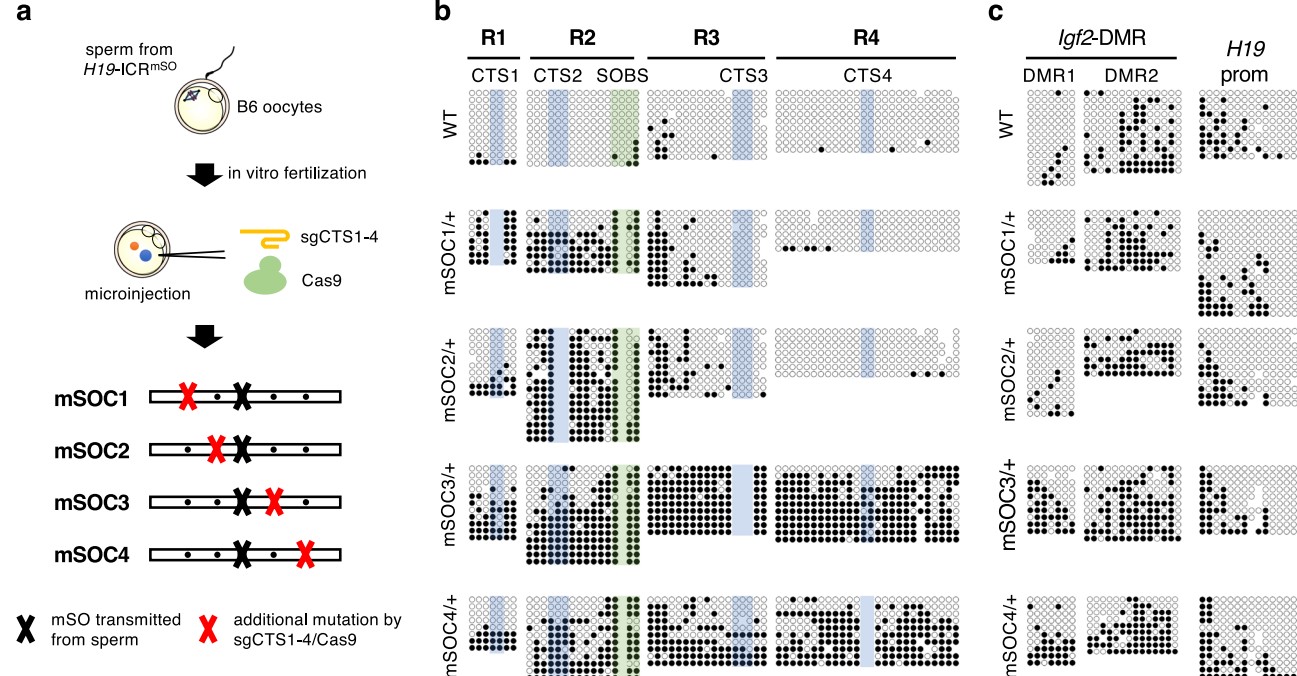

**Fig. 2 | DNA methylation status in SOBS/CTS double-mutant mice. a** Schematic representation of a protocol for the generation of SOBS/CTS double-mutant mice. **b** Methylation status at maternal *H19*-ICR in WT and *H19*-ICR^mSOC1-4/+ neonates. Representative results of maternal allele analysis are shown. Open and closed circles indicate unmethylated and methylated CpG sites, respectively. CpG sites included in CTS1-4 and SOBS are highlighted in blue and green, respectively. **c** Methylation status at maternal *Igf2*-DMR1, *Igf2*-DMR2, and *H19*prom in WT and *H19*-ICR^mSOC1-4/+ neonates. Representative results of maternal alleles are shown. Full results are shown in Supplementary Fig. 3.

microinjection of single-guide RNA (sgRNA)/Cas9/ssODN confirmed that the SOBS mutation in founder mice was consistent with the ssODN sequence, indicating the successful introduction of mutations in endogenous SOBS within *H19*-ICR (Supplementary Fig. 1). Neonates with a maternally inherited mSO allele (*H19*-ICR^mSO/+) and their wild-type (WT) littermates were obtained by crossing C57BL/6 J (B6)-background heterozygous female mice in the F1 generation with wild-type PWK/PhJ (PWK) male mice. The 2.6 kb region of *H19*-ICR dissected into four regions: R1, R2, R3, and R4 and included CTS1, CTS2, CTS3, and CTS4, respectively. R2 also included SOBS (Fig. 1a). Methylation levels were assessed through bisulfite sequencing to determine the methylation levels of the entire maternal *H19*-ICR. The total methylation levels of R1, R2, and R3 significantly increased in *H19*-ICR^mSO/+, whereas those of R4 did not (Fig. 1b and Supplementary Fig. 2a). CpGs around SOBS in R3 were hypermethylated, whereas CpGs in and around CTS3 were not (Fig. 1b). In other words, it is partial GOM.

We analyzed the expression levels of imprinted genes in the tissues of *H19*-ICR^mSO/+ and WT neonates to determine whether partial *H19*-ICR GOM caused the misexpression of imprinted genes in *H19*-ICR^mSO/+. Quantitative RT-PCR (qRT-PCR) analysis showed similar *Igf2* and *H19* levels in the liver, kidney, and tongue between *H19*-ICR^mSO/+ and WT (Fig. 1c). Allelic expression analysis confirmed the monoallelic expression of *Igf2* and *H19* in these tissues (Fig. 1d and Supplementary Fig. 2b). Consistent with these molecular analyses, *H19*-ICR^mSO/+ neonates were healthy and indistinguishable from their WT littermates. The body and liver weights of *H19*-ICR^mSO/+ neonates were not significantly different from those of WT neonates (Fig. 1e, f).

These results indicate that although maternal transmission of the SOBS mutation caused partial *H19*-ICR GOM, but did not cause imprinting defects in the *Igf2-H19* domain and BWS-like phenotypes. It is suggested that the absence of the BWS-like phenotype in *H19*-ICR^mSO/+ mice was due to the insufficient hypermethylation by a SOBS mutation, which does not extend beyond CTS3.

## Simultaneous mutation of SOBS and CTS causes *H19*-ICR GOM

Since CTCF binding to CTSs is essential for the unmethylated state of the maternal *H19*-ICR, we hypothesized that the extension of hypermethylation via SOBS mutation is prevented by the CTCF binding to CTSs (i.e., CTS3 and CTS4) in *H19*-ICR^mSO/+ mice. To this end, we used the CRISPR/Cas9 system to introduce additional mutations to CTS3 or CTS4 regions of the *H19*-ICR^mSO alleles (namely *H19*-ICR^mSOC3 and *H19*-ICR^mSOC4) (Fig. 2a). In addition, we also generated mice with mutations at CTS1 or CTS2 on the *H19*-ICR^mSO alleles, respectively (namely *H19*-ICR^mSOC1 and *H19*-ICR^mSOC2). Bisulfite sequencing of *H19*-ICR in neonates of *H19*-ICR^mSOC3/+ and *H19*-ICR^mSOC4/+ showed hypermethylation of R3 and R4 in addition to the regions hypermethylated in *H19*-ICR^mSO/+ mice (Fig. 2b and Supplementary Fig. 3). Methylation levels of R3 in the maternal allele of *H19*-ICR^mSOC3/+ were significantly higher than those of the maternal allele of *H19*-ICR^mSOC4/+ (*H19*-ICR^mSOC3/+ vs. *H19*-ICR^mSOC4/+; 94% vs. 50%, respectively, *P* < 0.0001 (Mann-Whitney U test)). In *H19*-ICR^mSOC1/+, although the region from R1 to CpGs around the mutated SOBS was hypermethylated, CpGs around CTS3 and R4 remained unmethylated as with *H19*-ICR^mSO/+. Methylation status of *H19*-ICR^mSOC2/+ was similar to that of *H19*-ICR^mSOC1/+, except for the higher methylation in R2 (Fig. 2b and Supplementary Fig. 3).

Next, we evaluated DNA methylation levels at *Igf2*-DMR1, *Igf2*-DMR2, and *H19*prom to determine whether *H19*-ICR GOM in these mutant mice affects the DNA methylation status of somatic DMRs in the *Igf2-H19* domain. The methylation levels at *Igf2*-DMR1 and *H19*prom on the maternal allele significantly increased in *H19*-ICR^mSOC3/+ and *H19*-ICR^mSOC4/+ compared to WT. In contrast, those in the maternal *Igf2*-DMR2 were not significantly different among all mutant mice and WT (Fig. 2c and Supplementary Fig. 3). In *H19*-ICR^mSOC1/+ and *H19*-ICR^mSOC2/+, the methylation levels at all these somatic DMRs were not different compared to WT.

We further generated mice with single mutations at each CTS1, CTS2, CTS3, and CTS4 (*H19*-ICR^ΔC1, *H19*-ICR^ΔC2, *H19*-ICR^ΔC3, and *H19*-ICR^ΔC4, respectively) to elucidate the effects of CTS mutations on maternal *H19*-ICR

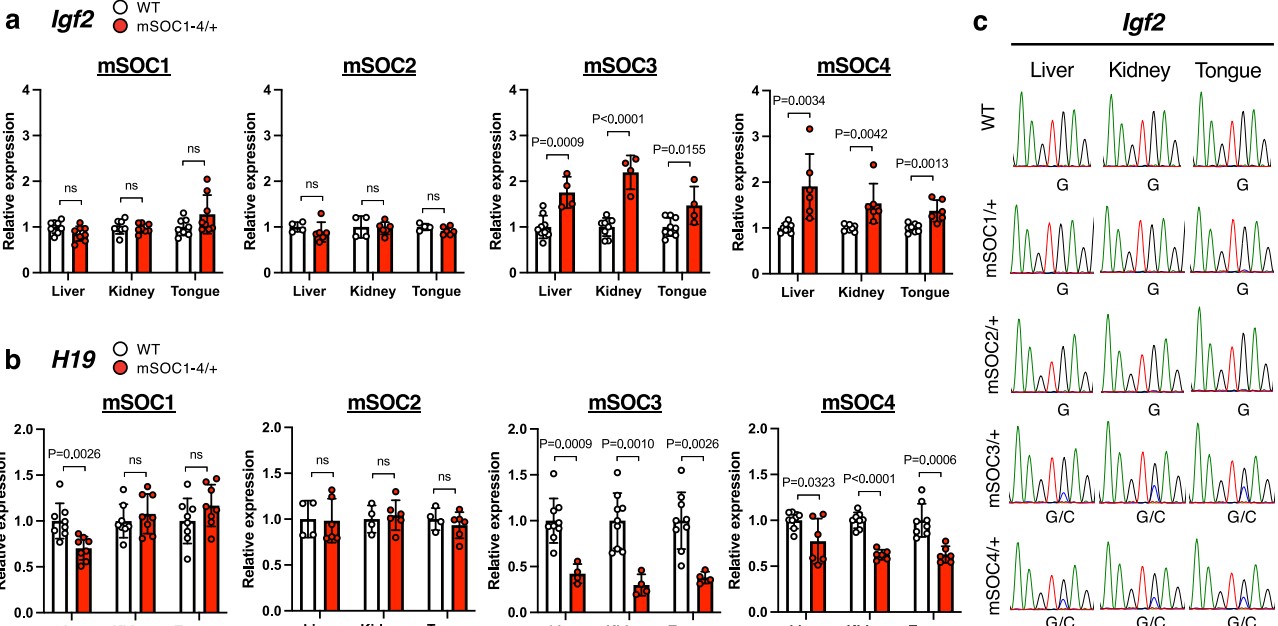

**Fig. 3 | Expression of imprinted genes in SOBS/CTS double-mutant mice.**
Expression levels of *Igf2* (**a**) and *H19* (**b**) in *H19*-ICR^mSOC1/+ (WT and *H19*-ICR^mSOC1/+; *n* = 8 and *n* = 8, respectively, from 2 litters), *H19*-ICR^mSOC2/+ (WT and *H19*-ICR^mSOC2/+; *n* = 4 and *n* = 6, respectively, from 2 litters), *H19*-ICR^mSOC3/+ (WT and *H19*-ICR^mSOC3/+; *n* = 9 and *n* = 4, respectively, from 3 litters), and *H19*-ICR^mSOC4/+ (WT and *H19*-ICR^mSOC4/+; *n* = 8 and *n* = 6, respectively, from 3 litters) tissues. White and red bars indicate WT and *H19*-ICR^mSOC1-4/+ tissues, respectively. Error bars indicate standard deviation. *P*-values are indicated (unpaired two-tailed *t*-test). ns; not significant. **c** Allelic expression of *Igf2* in WT and *H19*-ICR^mSOC1-4/+ neonates. Representative electropherograms of the RT-PCR products are shown.

alleles. The methylation level of each mutated site significantly increased in mutant neonates, whereas other regions remained unmethylated (Supplementary Fig. 4). These results suggest that a single mutation in CTS1-4 has little effect on maternal *H19*-ICR except for each mutated site. In addition, no methylation changes were observed at *Igf2*-DMR1, *Igf2*-DMR2, and *H19*prom in these mutants (Supplementary Fig. 4).

These results indicate that SOBS, CTS3, and CTS4 are important for GOM of the entire *H19*-ICR and somatic DMRs within *Igf2*-*H19* domain.

## Double mutation of SOBS/CTS on maternal *H19*-ICR impairs the imprinted expression of *Igf2* and *H19*

To investigate whether *H19*-ICR GOM in SOBS/CTS mutants causes imprinting defects in the *Igf2*-*H19* domain, we analyzed the expression levels of imprinted genes in liver, kidney, and tongue of 1 day postpartum (dpp) neonates from *H19*-ICR^mSOC1-4/+ and *H19*-ICR^ΔC1-4/+. The expression of *Igf2* was significantly upregulated in all analyzed tissues of *H19*-ICR^mSOC3/+ and *H19*-ICR^mSOC4/+ compared to their WT littermates (Fig. 3a). The degrees of *Igf2* elevation in liver and tongue were comparable between *H19*-ICR^mSOC3/+ and *H19*-ICR^mSOC4/+ (*H19*-ICR^mSOC3/+ vs. *H19*-ICR^mSOC4/+; 1.7- vs. 1.9-fold in the liver and 1.5- vs. 1.3-fold in the tongue, respectively) (Fig. 3a). However, that in kidney were higher in *H19*-ICR^mSOC3/+ than in *H19*-ICR^mSOC4/+ (2.1- vs. 1.5-fold, respectively). Moreover, no changes in *Igf2* expression were observed in all tissues of *H19*-ICR^mSOC1/+ and *H19*-ICR^mSOC2/+.

We also analyzed the expression levels of maternally expressed *H19*. *H19* expression was significantly reduced in all tissues of *H19*-ICR^mSOC3/+ and *H19*-ICR^mSOC4/+. The degrees of reduction were greater in *H19*-ICR^mSOC3/+ tissues than in *H19*-ICR^mSOC4/+ tissues (*H19*-ICR^mSOC3/+ vs. *H19*-ICR^mSOC4/+; 0.4- vs. 0.7-fold in the liver; 0.3- vs. 0.6-fold in the kidney; 0.4- vs. 0.6-fold in the tongue, respectively). In *H19*-ICR^mSOC1/+ and *H19*-ICR^mSOC2/+, *H19* expression in all tissues was not different compared to WT except for the liver of *H19*-ICR^mSOC1/+ (0.7-fold, *P* < 0.001) (Fig. 3b). In addition, the expression levels of *Igf2* and *H19* in *H19*-ICR^ΔC1-4/+ were not significantly different from those in WT (Supplementary Fig. 5a).

To investigate whether the *Igf2* upregulation resulted from the biallelic expression, the allelic expression in *H19*-ICR^mSOC3/+ and *H19*-ICR^mSOC4/+ tissues was analyzed by detecting polymorphisms between B6 and PWK. Biallelic expression of *Igf2* was observed in *H19*-ICR^mSOC3/+ and *H19*-ICR^mSOC4/+ but not in *H19*-ICR^mSOC1/+ and *H19*-ICR^mSOC2/+ (Fig. 3c). Allelic expression analysis confirmed monoallelic expression of *Igf2* in all tissues of *H19*-ICR^ΔC1-4/+ (Supplementary Fig. 5b).

These results indicate that *H19*-ICR GOM observed in *H19*-ICR^mSOC3/+ and *H19*-ICR^mSOC4/+ neonates is sufficient for imprinting defects at *Igf2*-*H19* domain, and that the mutation of a single CTS does not cause misexpression of imprinted genes.

## *H19*-ICR GOM by SOBS/CTS mutation causes BWS-like overgrowth

We analyzed the body and tissue weight of *H19*-ICR^mSOC1-4/+ neonates to determine whether the alteration of gene expression in SOBS/CTS double mutants causes BWS-like phenotypes. The body weights of *H19*-ICR^mSOC3/+ and *H19*-ICR^mSOC4/+ at 1 dpp were significantly higher than those of their WT littermates (Fig. 4a). Contrastingly, the body weights of *H19*-ICR^mSOC1/+ and *H19*-ICR^mSOC2/+ neonates were not significantly different. *H19*-ICR^mSOC3/+ neonates exhibited increased liver and tongue weights compared with WT neonates. *H19*-ICR^mSOC4/+ neonates also showed increased tongue weight, however, liver weight was not different (Fig. 4b, c). In *H19*-ICR^mSOC1/+ and *H19*-ICR^mSOC2/+ neonates, liver and tongue weights were not significantly different (Fig. 4b, c). In addition, kidney weight was not significantly different in *H19*-ICR^mSOC1-4/+ neonates (Supplementary Fig. 6a). Moreover, the body and tissue weights in *H19*-ICR^ΔC1-4/+ neonates were indistinguishable from those of their WT littermates (Supplementary Fig. 6b–e).

We next focused on the phenotypic differences between *H19*-ICR^mSOC3/+ and *H19*-ICR^mSOC4/+ neonates. We plotted the weights of neonates from birth until weaning to determine the postnatal growth. *H19*-ICR^mSOC3/+ neonates were significantly larger than their WT littermates throughout the postnatal period (Fig. 5a). However, the weight of *H19*-ICR^mSOC4/+ neonates did not significantly differ from that of WT

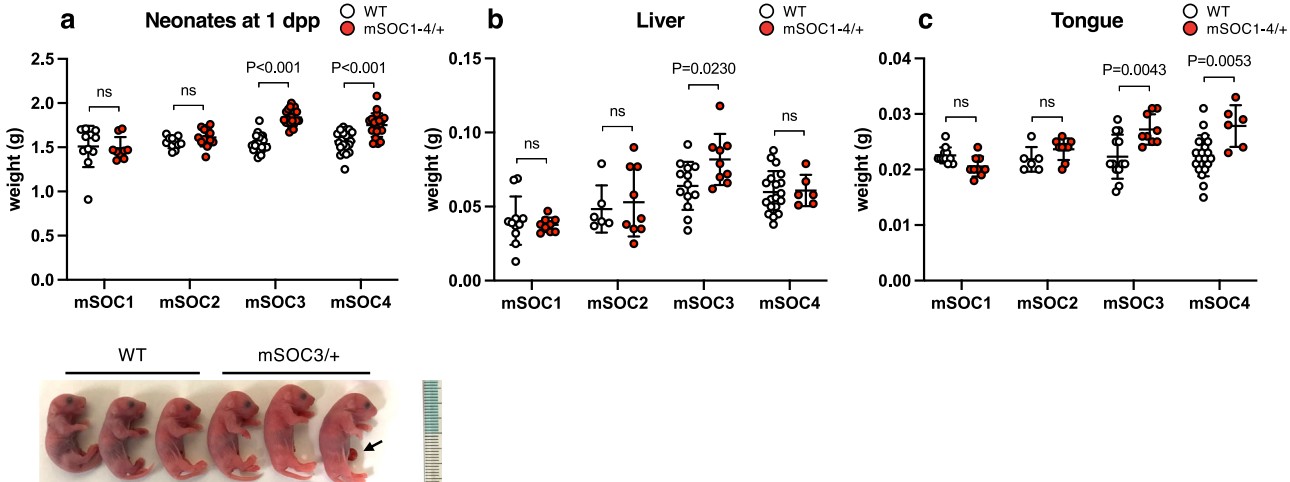

**Fig. 4 | The phenotypes of SOBS/CTS double-mutant mice. a** (Top) Body weights of neonates at 1 dpp in *H19*-ICR[mSOC1/+] (WT and *H19*-ICR[mSOC1/+]; *n* = 11 and *n* = 9, respectively, from 3 litters), *H19*-ICR[mSOC2/+] (WT and *H19*-ICR[mSOC2/+]; *n* = 13 and *n* = 11, respectively, from 3 litters), *H19*-ICR[mSOC3/+] (WT and *H19*-ICR[mSOC3/+]; *n* = 26 and *n* = 19, respectively, from 5 litters), and *H19*-ICR[mSOC4/+] (WT and *H19*-ICR[mSOC4/+]; *n* = 24 and *n* = 20, respectively, from 5 litters). White and red dots indicate individual weights of WT and heterozygous mutants with maternally transmitted *H19*-ICR[mSOC1-4] alleles, respectively. Horizontal and error bars indicate the mean ± SD, respectively.

(bottom) Representative photographs of *H19*-ICR[mSOC3/+] neonates and WT littermates. The arrow indicates umbilical hernia. The liver (**b**) and tongue (**c**) weights of neonates at 1 dpp: *H19*-ICR[mSOC1/+] (WT and *H19*-ICR[mSOC1/+]; *n* = 11 and *n* = 9, respectively, from 3 litters), *H19*-ICR[mSOC2/+] (WT and *H19*-ICR[mSOC2/+]; *n* = 6 and *n* = 9, respectively, from 2 litters), *H19*-ICR[mSOC3/+] (WT and *H19*-ICR[mSOC3/+]; *n* = 9 and *n* = 13, respectively, from 3 litters), and *H19*-ICR[mSOC4/+] (WT and *H19*-ICR[mSOC4/+]; *n* = 19 and *n* = 6, respectively, from 3 litters). *P*-values are indicated (unpaired two-tailed *t*-test). ns; not significant.

mice after 9 dpp except for 13 dpp (Fig. 5b). Moreover, the frequency of overgrowth in *H19*-ICR[mSOC3/+] neonates ( ≥ mean + 2 SD of WT littermate) was higher than that in *H19*-ICR[mSOC4/+] neonates despite there being no significant difference (*H19*-ICR[mSOC3/+] vs. *H19*-ICR[mSOC4/+]; 79% (15 out of 19) vs. 50% (10 out of 20); *P* = 0.0958, Fisher's exact test).

We analyzed the weights of embryos and placentas to investigate whether the SOBS/CTS mutation affects embryonic and placental development. *H19*-ICR[mSOC3/+] mice had increased fetal and placental weights from 16.5 days postcoitum (dpc), whereas neither were significantly higher in *H19*-ICR[mSOC4/+] mice than in WT mice (Fig. 5c–f).

Hypomethylation of paternal *H19*-ICR causes imprinting defects of the *Igf2-H19* domain, resulting in SRS-like growth retardation in mice[22,23]. To investigate the effect of paternal transmission of the mutated allele, we generated mice with paternally transmitted *H19*-ICR[mSOC3] allele (*H19*-ICR[+/mSOC3]) by crossing *H19*-ICR[mSOC3] male mice with PWK females. DNA methylation analysis showed that *H19*-ICR on the paternal allele was completely methylated in *H19*-ICR[+/mSOC3] neonates (Supplementary Fig. 7a). Moreover, no altered expression of the imprinted genes was observed in *H19*-ICR[+/mSOC3] tissues (Supplementary Fig. 7b and c). Consistent with DNA methylation and gene expression, *H19*-ICR[mSOC3/+] neonates were indistinguishable from their WT littermates and did not show any SRS-like phenotypes (Supplementary Fig. 7d–f). These results indicate that SOBS and CTS3 mutations do not cause imprinting defects at paternal *H19*-ICR.

Taken together, our results indicate that simultaneous mutation of SOBS/CTS3 or SOBS/CTS4, but not that of SOBS/CTS1 or SOBS/CTS2, on the maternal allele results in BWS-like overgrowth. Moreover, simultaneous mutations in CTS3 and SOBS caused more severe BWS-like phenotypes than those in CTS4 and SOBS.

## DNA methylation dynamics of maternal *H19*-ICR in SOBS/CTS mutants

We assessed the DNA methylation status of R2 and R3 in oocytes, blastocysts, and embryos to determine when aberrant hypermethylation of maternal *H19*-ICR occurred. In WT and *H19*-ICR[mSO/+], R2 and R3 of the maternal allele were hypomethylated in oocytes and blastocysts, whereas DNA methylation levels significantly increased in embryos at 6.5 dpc,

consistent with previous reports[21,24] (Supplementary Fig. 8). In *H19*-ICR[mSOC3/+], R2 and R3 of the maternal allele was also unmethylated in oocytes and blastocysts, whereas it was highly methylated especially in R3 at 6.5 dpc (Supplementary Fig. 9). These results indicate that *H19*-ICR GOM, which causes imprinting defect and BWS-like overgrowth, occurs during post-implantation period as same as WT and *H19*-ICR[mSO/+].

## Simultaneous mutation of CTS3/CTS4 partially recapitulates the phenotypes of SOBS/CTS3 mutation

We generated CTS3/CTS4 double mutant mice (*H19*-ICR[ΔC3,4]) using CRISPR/Cas9 to determine whether SOBS is sufficient for the unmethylated state of maternal *H19*-ICR (Fig. 6a). The region including SOBS, CTS3, and CTS4 was hypermethylated in *H19*-ICR[ΔC3,4/+] neonates. In contrast, the 5′ regions flanking SOBS, CTS1, and CTS2 were only partially methylated (Fig. 6b and Supplementary Fig. 10a). In addition, maternal *Igf2*-DMR1 was hypermethylated but the methylation levels of *Igf2*-DMR2 and *H19*prom were not significantly different compared to those of their WT littermates (Fig. 6c and Supplementary Fig. 10a). Furthermore, at 6.5 dpc, a period when OCT4 expression peaks, four CpGs within SOBS remained hypomethylated, whereas R3 was highly methylated in *H19*-ICR[ΔC3,4/+] (Supplementary Fig. 10b).

We evaluated the expression levels of imprinted genes in *H19*-ICR[ΔC3,4/+] neonatal tissues. Expression levels were significantly increased, owing to biallelic expression of *Igf2*, in all *H19*-ICR[ΔC3,4/+] tissues (Fig. 6d, e). In addition, the elevated *Igf2* levels were comparable to those of *H19*-ICR[mSOC4/+] neonates, but not to those of *H19*-ICR[mSOC3/+] neonates (Supplementary Fig. 11). In contrast, *H19* expression was substantially reduced in the liver, but it was slight in other tissues (Fig. 6d).

Phenotypic analysis of *H19*-ICR[ΔC3,4/+] neonates showed that 18% (3 out of 17) of *H19*-ICR[ΔC3,4/+] neonates exhibited BWS-like overgrowth, which was significantly less frequent than in *H19*-ICR[mSOC3/+] neonates (*H19*-ICR[ΔC3,4/+] vs. *H19*-ICR[mSOC3/+]; 18% vs. 79%, *P* = 0.0008, Fisher's exact test). Although the average body weight was not significantly different from that of the WT littermates (Fig. 6f), the weights of liver and tongue in *H19*-ICR[ΔC3,4/+] were significantly increased compared to those in WT (Fig. 6g, h). No significant differences in fetal body weight were observed. However, placental weight significantly increased at 15.5 and 16.5 dpc

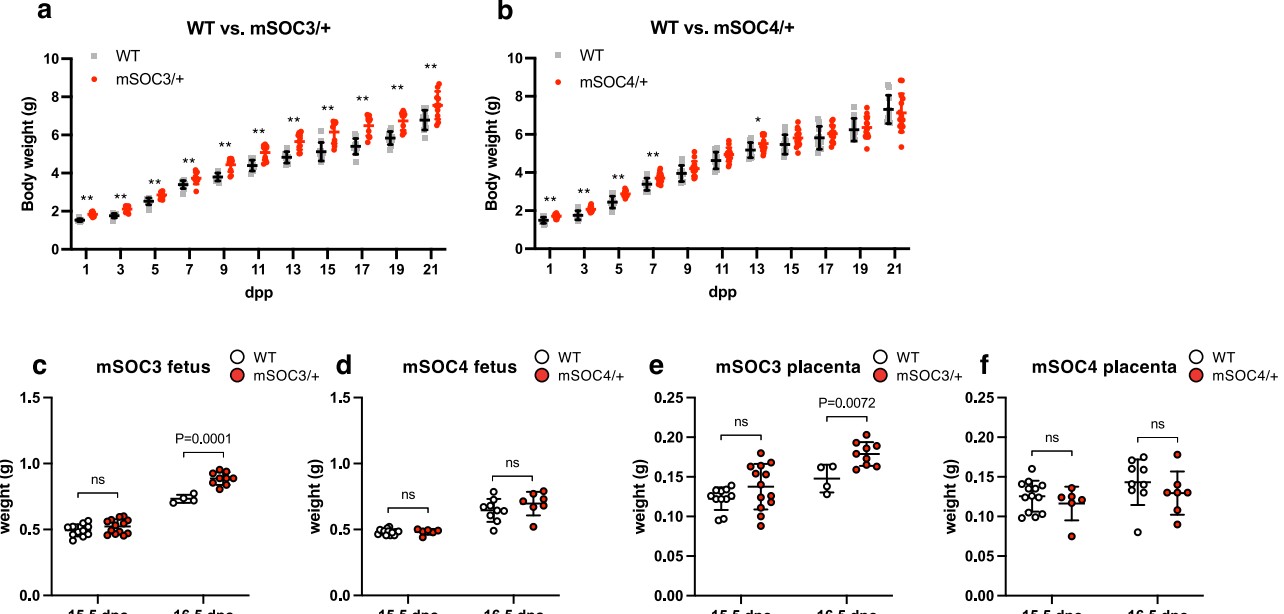

**Fig. 5 | The phenotypes of *H19*-ICR^mSOC3/+ and *H19*-ICR^mSOC4/+ mice. a** Growth of *H19*-ICR^mSOC3/+ (*n* = 10) and WT (*n* = 10), as well as (**b**) *H19*-ICR^mSOC4/+ (*n* = 14) and WT (*n* = 10) neonates from 3 litters. Gray squares and red dots indicate individual weights of WT and heterozygous mutants with maternally transmitted *H19*-ICR^mSOC3 or *H19*-ICR^mSOC4 alleles, respectively. Horizontal and error bars indicate the mean ± SD, respectively. **; *P* < 0.01, *; *P* < 0.05 (two-tailed unpaired *t*-test). **c** Fetal weights in WT (*n* = 11) and *H19*-ICR ^mSOC3/+ (*n* = 13) mice from 3 litters at 15.5 dpc and WT (*n* = 4) and *H19*-ICR^mSOC3/+ (*n* = 9) from 2 litters at 16.5 dpc. **d** Fetal weights in WT (*n* = 13) and *H19*-ICR ^mSOC4/+ (*n* = 6) mice from 3 litters

at 15.5 dpc and WT (*n* = 9) and *H19*-ICR ^mSOC4/+ (*n* = 7) from 3 litters at 16.5 dpc. **e** Placental weights in WT (*n* = 11) and *H19*-ICR^mSOC3/+ (*n* = 13) mice from 3 litters at 15.5 dpc and WT (*n* = 4) and *H19*-ICR ^mSOC3/+ (*n* = 9) from 2 litters at 16.5 dpc. **f** Placental weights in WT (*n* = 13) and *H19*-ICR ^mSOC4/+ (*n* = 6) mice from 3 litters at 15.5 dpc and WT (*n* = 9) and *H19*-ICR ^mSOC4/+ (*n* = 7) from 3 litters at 16.5 dpc. White and red dots indicate individual weights of WT and heterozygous mutants with maternally transmitted *H19*-ICR^mSOC3 or *H19*-ICR^mSOC4 alleles, respectively. Horizontal and error bars indicate the mean ± SD, respectively. *P*-values are indicated (unpaired two-tailed *t*-test). ns; not significant.

(Fig. 6i, j). These results suggest that SOBS itself is not sufficient to maintain the unmethylated state of *H19*-ICR, and that CTS3/CTS4 mutations cause partial GOM of *H19*-ICR and misexpression of imprinted genes, but the effect on BWS-like overgrowth was not as strong as that of the SOBS/CTS3 mutation.

Taken together, our findings indicate that SOBS and CTS3 collaborate to maintain the unmethylated state of maternal *H19*-ICR and that mutations of SOBS and CTS3 cause *H19*-ICR GOM-induced BWS-like overgrowth in mice.

### Cohesin binding to CTS3 is essential for maternal *H19*-ICR

As the binding of CTCF and the cohesin complex to *H19*-ICR is essential for the chromatin architecture of the maternal *Igf2-H19* domain, we conducted chromatin immunoprecipitation (ChIP) analysis at CTS1-4 in SOBS/CTS and CTS mutants. To this end, chromatin samples of embryos at 10.5 dpc were immunoprecipitated using antibodies against CTCF and RAD21, a cohesin subunit that facilitates cohesin loading and CTCF/cohesin-mediated 3D genome organization[25]. The enrichment of CTCF and RAD21 at CTS1-4 in *H19*-ICR^ΔC3/+, *H19*-ICR^ΔC4/+, and *H19*-ICR^ΔC3,4/+ did not change compared to the WT, except for mutated CTS (Fig. 7, Supplementary Fig. 12). Importantly, the enrichment of CTCF at CTS1, CTS2, and CTS4 significantly decreased in *H19*-ICR^mSOC3/+, corresponding with the DNA methylation status of the maternal allele (Fig. 7a). The enrichment of RAD21 at CTS2 and CTS4, but not CTS1, significantly decreased (Fig. 7b). In *H19*-ICR^mSOC4/+, the enrichment of CTCF at CTS1 did not change, but those at CTS2 and CTS3 were significantly but decreased to a smaller extent. However, the enrichment of RAD21 at CTS1, CTS2, and CTS3 did not change. These results suggest that the binding of both CTCF and the cohesin complex to maternal *H19*-ICR is disrupted in *H19*-ICR^mSOC3/+ embryos. Overall, these results suggest that the binding of CTCF and the cohesin complex to CTS3 is the primary contributor to the formation of a maternal-specific chromatin state.

## Discussion

This study demonstrates that SOBS and CTS3 collaborate to maintain the unmethylated state of maternal *H19*-ICR and are responsible sequences for *H19*-ICR GOM leading to imprinting defects and BWS-like overgrowths. Furthermore, CTS3 bound by CTCF/cohesin is critical for the maternal-specific chromatin structure. Our comprehensive methylation analysis revealed that SOBS mutation (*H19*-ICR^mSO/+) can cause partial *H19*-ICR GOM (Fig. 8a). However, its effects were prevented shortly before CTS3, suggesting that CTS3 functions as a "DNA methylation boundary"[26,27]. The CTCF-bound DNA methylation boundary that separates the unmethylated promoter region and the hypermethylated intergenic region was reported to be located 5′ upstream of human *FMR1*[27]. In fragile X syndrome (FXS) patients with CGG triplet repeat expansion, the promoter region of *FMR1* was aberrantly hypermethylated beyond the boundary and the CTCF binding to the boundary was lost[28]. Moreover, the disruption of a topologically associated domain (TAD) boundary identified near the DNA methylation boundary correlates with *FMR1* silencing in patients with FXS due to hypermethylation of expanded CGG triplet repeat. This suggests that chromatin interactions through CTCF/cohesin binding may simultaneously define the TAD and DNA methylation boundary[29]. Similarly, the binding of CTCF to maternal *H19*-ICR mediates allele-specific TAD formation in mouse embryonic stem cells[30]. These suggest that failure of CTCF binding to CTS3 concomitantly disrupts the maternal-specific TAD structure and DNA methylation boundary in *H19*-ICR^mSOC3/+ mice, resulting in *H19*-ICR GOM. Mice generated by Ideraabdullah et al. (*H19*^ΔC2,3/+) harboring a deletion including SOBS, CTS2, and CTS3 showed unmethylated status for the remaining CTSs on the maternal allele (CTS1 and CTS4)[23] and smaller effects than *H19*-ICR^mSOC3/+. Since CTS spacing was reported to alter CTCF-dependent insulation activity, a deletion in *H19*^ΔC2,3/+ might alter the spacing between CTS1 and CTS4, and rescue maternal-specific TAD formation

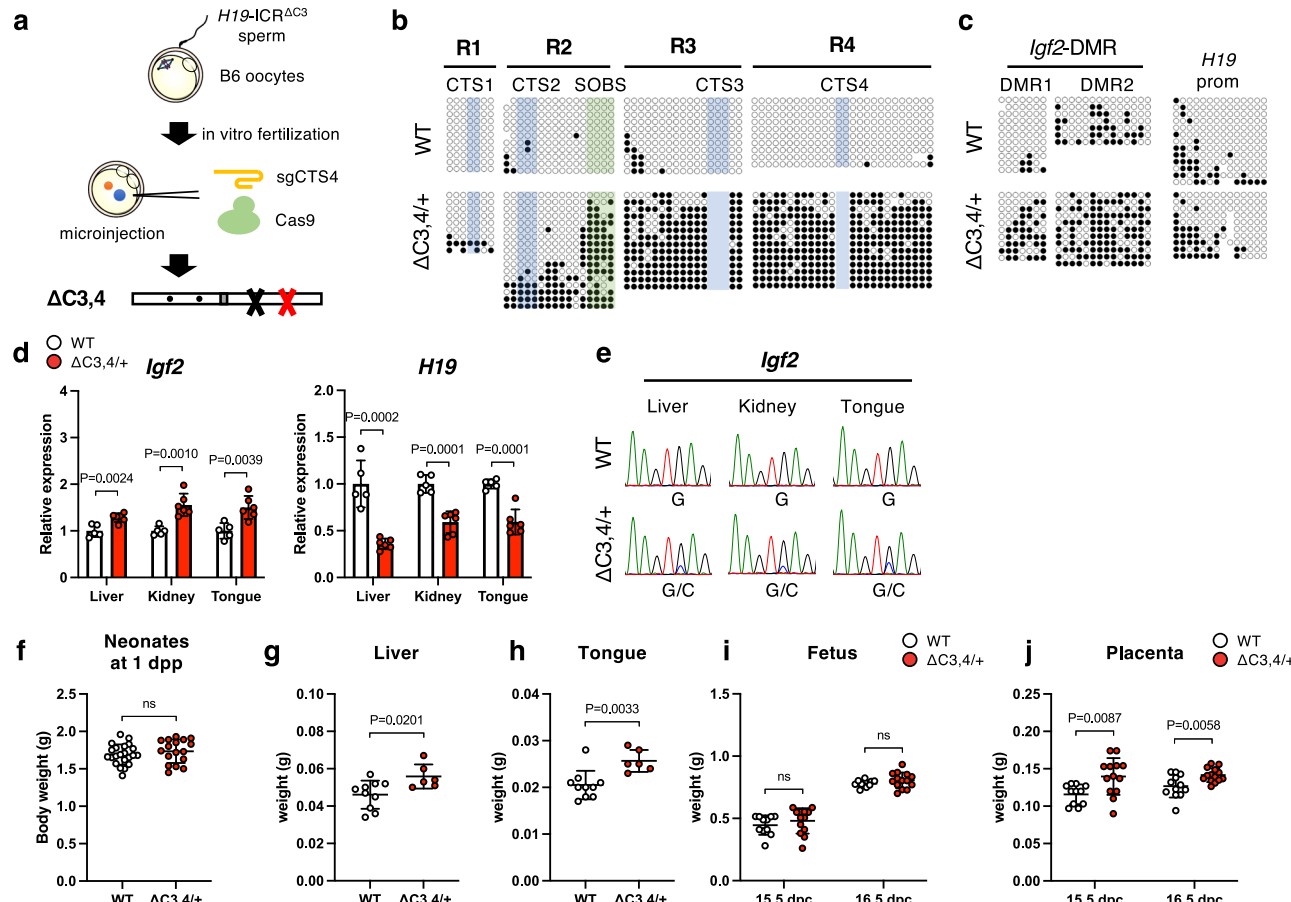

**Fig. 6 | The phenotypes of *H19*-ICR^ΔC3,4/+ mice. a** Schematic representation of a protocol for the generation of SOBS/CTS double-mutant mice. **b** Methylation status at maternal *H19*-ICR in WT and *H19*-ICR^ΔC3,4/+ neonates. Representative results of maternal alleles are shown. Full results are shown in Supplementary Fig. 9a. **c** Methylation status at maternal *Igf2*-DMR1, *Igf2*-DMR2, and *H19*prom in WT and *H19*-ICR^ΔC3,4/+ neonates. Representative results of maternal alleles are shown. Full results are shown in Supplementary Fig. 9b. **d** Expression levels of *Igf2* and *H19* in WT ($n = 5$) and *H19*-ICR^ΔC3,4/+ ($n = 6$) tissues from 2 litters. White and red bars indicate WT and mutant tissues, respectively. Error bars indicate standard deviation. **e** Allelic expression of *Igf2* in WT and *H19*-ICR^ΔC3,4/+ tissues. Representative electropherograms of the RT-PCR products are shown. **f** Body weights of WT ($n = 22$) and *H19*-ICR^ΔC3,4/+ ($n = 17$) tissues from 6 litters at 1 dpp. Horizontal and error bars indicate the mean ± SD, respectively. Liver (**g**) and tongue (**h**) weights in WT ($n = 10$) and *H19*-ICR^ΔC3,4/+ ($n = 6$) tissues from 2 litters at 1 dpp. **i** Fetal weights in WT ($n = 11$) and *H19*-ICR^ΔC3,4/+ ($n = 13$) mice from 3 litters at 15.5 dpc and WT ($n = 12$) and *H19*-ICR^ΔC3,4/+ ($n = 14$) mice from 3 litters at 16.5 dpc. **j** Placental weights in WT ($n = 11$) and *H19*-ICR^ΔC3,4/+ ($n = 13$) mice from 3 litters at 15.5 dpc and WT ($n = 12$) and *H19*-ICR^ΔC3,4/+ ($n = 14$) from 3 litters at 16.5 dpc. White and red dots indicate individual weights of WT and *H19*-ICR^ΔC3,4/+, respectively. Horizontal and error bars indicate the mean ± SD, respectively. *P*-values are indicated (unpaired two-tailed *t*-test). ns; not significant.

in *Igf2-H19*[31]. Moreover, we found that *H19*-ICR^ΔC3/+ neonates showed partial maternal *H19*-ICR GOM and did not exhibit any imprinting defects, suggesting that the maternal-specific TAD structure including *Igf2-H19* is rescued by CTCF binding on remaining CTSs in *H19*-ICR^ΔC3/+ and that the maternal-specific TAD limits the expansion of *H19*-ICR GOM. In addition, SOBS was highly methylated in *H19*-ICR^ΔC3,4/+ neonates, but with little spreading toward CTS1 and CTS2. This implies that another methylation boundary might exist between SOBS and CTS2. Furthermore, CTCF-dependent genome-wide TAD organization is established during preimplantation development, suggesting that the TAD and DNA methylation boundary of maternal *H19*-ICR mediated by SOBS and CTS3 might be established until blastocyst stage[32].

Transgenic mice have shown that the excess expression of *Igf2* causes tissue overgrowth in a dose-dependent manner, suggesting that neonatal tissue overgrowth depends only upon *Igf2*[33]. However, our data showed that the overgrowth of liver, tongue, placenta in *H19*-ICR^mSOC3/+, *H19*-ICR^mSOC4/+, and *H19*-ICR^ΔC3,4/+ neonates, except for liver and placenta in *H19*-ICR^mSOC4/+ neonates, was associated not only with the increased expression of *Igf2* but also with the decreased expression of *H19* (Fig. 8a). Dependence of overgrowth on the decrease in *H19*

expression is supported by a report by Park et al. in which adult cardiac hypertrophy and fibrosis caused by loss of imprinting at *Igf2-H19* locus were restored by *H19* expression using a bacterial artificial chromosome transgene[34]. As for the kidney in these mutants, overgrowth was not observed despite both the increased *Igf2* expression and the decreased *H19* expression. Since increased kidney weight has been observed only in transgenic lines with highly expressing *Igf2*[33], the dosage of *Igf2* in the kidney of our mutant mice is not sufficient to induce kidney overgrowth. In addition, the liver in *H19*-ICR^mSOC1/+, in which only *H19* was decreased but did not change CTCF/cohesin enrichment, overgrowth was not observed. These suggests that a tissue-specific chromatin organization exists and regulates tissue-dependent insulation activity and that tissue-specific threshold for the levels of *Igf2* and *H19* determines tissue weight.

Consistent with previous reports[15,24], our results showed that *H19*-ICR GOM occurred during post-implantation development in mutant mice. As DNA methylation after implantation depends on DNA methyltransferase 3B (DNMT3B)[34], the binding of OCT4 and CTCF to SOBS and CTS3 on the maternal allele may block the access of DNMT3B at this stage. Moreover, global 5-hydroxymethylcytosine levels increase in post-implantation embryos[35]. Pioneering transcription factors, including OCT4 and SOX2,

## a  CTCF ChIP

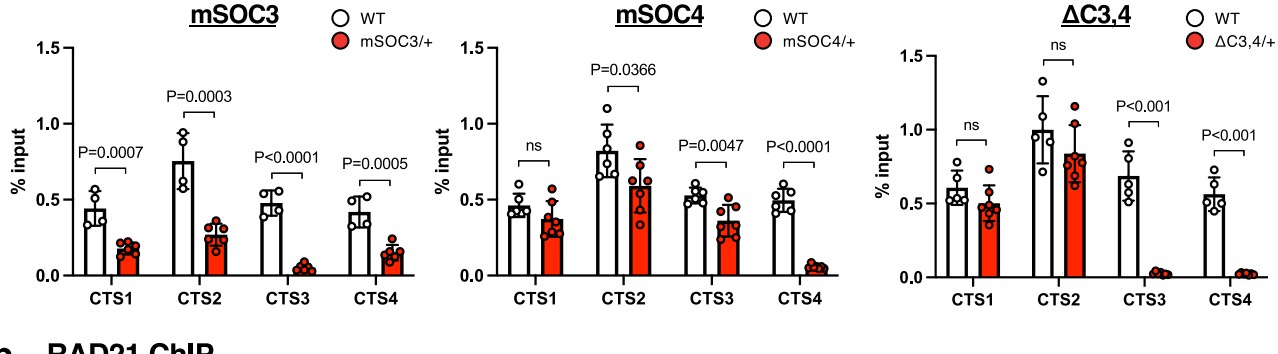

## b  RAD21 ChIP

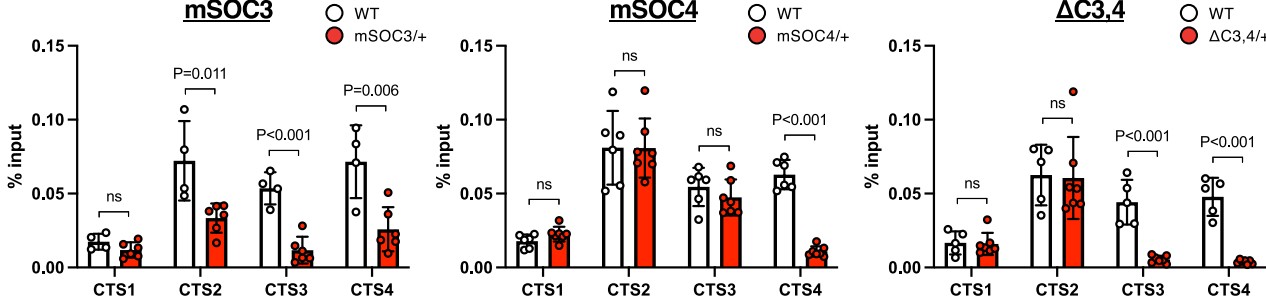

**Fig. 7 | Enrichment of CTCF and RAD21 in SOBS/CTS double-mutant embryos.** ChIP-qPCR analysis of CTS1-4 in *H19*-ICR[mSOC3/+] (*n* = 6) and WT (*n* = 4), *H19*-ICR[mSOC4/+] (*n* = 7) and WT (n = 6), and *H19*-ICR[ΔC3,4/+] (*n* = 7) and WT (*n* = 5) embryos from 2 litters at 10.5 dpc, as performed using CTCF (**a**) and RAD21 (**b**) antibodies. White and red bars indicate the WT and mutant, respectively. Error bars indicate standard deviation. *P*-values are indicated (unpaired two-tailed *t*-test). ns; not significant.

can protect their binding sites from DNA methylation by inducing TET-dependent active DNA demethylation[36,37]. Therefore, TET family proteins may be recruited by OCT4 binding to maternal SOBS in *H19*-ICR. According to the ChIP-Atlas website (https://chip-atlas.org/peak_browser), other transcription factors or epigenetic regulators also bind around SOBS in mouse ES cells, in addition to SOX2 and OCT4. This implies that the binding of these factors might be involved in maintaining the unmethylated state of maternal *H19*-ICR; however, the detailed molecular mechanism of this binding is largely unknown. Further studies are required to elucidate this mechanism.

Our results indicate that the frequency of overgrowth was different among various mutant mice, and the severity of overgrowth was different in individuals harboring the same genotype. The variety of phenotypes observed in patients with BWS is attributable to mosaicism between normal and *H19*-ICR GOM cells[8]. A variety of DNA methylation levels at maternal *H19*-ICR and *Igf2*-DMR1 among the mutant mice with the same genotype (Supplementary Figs. 3b–c and 4b–c) suggests mosaicism in normal and *H19*-ICR GOM cells in the mutant mice as well as in humans. Thus, stochastic epigenetic mechanisms may be responsible for the pathogenesis of overgrowth in mutant mice.

We found that SOBS and CTS3 are the responsible sequences in which mutations cause *H19*-ICR GOM leading to BWS-like overgrowth in mice. However, several differences exist between mice and humans. In mice, a point mutation in SOBS causes partial *H19*-ICR GOM but not overgrowth. In contrast, it is sufficient for both GOM of entire *H19*-ICR and BWS phenotypes in humans[16–18]. By using humanized mice, a positional relationship between SOBS and CTS has been reported to be similar between humans and mice[38–40]. However, the conserved region upstream of *H19*-ICR plays a central role in chromatin organization in the human *Igf2-H19* domain, but not in mice[41,42]. This implies a difference in the establishment of maternal *H19*-ICR chromatin structure between humans and mice.

Moreover, recent studies have reported that a point mutation in the OCT4-binding site in human *H19*-ICR causes anticipation of DNA methylation, wherein the DNA methylation levels at *H19*-ICR gradually increase over two generations[43,44]. However, this phenomenon was not observed in our mutant mice at least in two generations.

In summary, we have identified SOBS and CTS3 as the responsible sequences for protecting the unmethylated state of maternal *H19*-ICR, and their mutations cause *H19*-ICR GOM, leading to BWS-like overgrowth. *H19*-GOM with BWS-like overgrowth in mice. Our results indicate that the binding of OCT4 and CTCF/cohesin to SOBS and CTS3, respectively, may define the putative DNA methylation boundary and maintain the unmethylated state of maternal *H19*-ICR in mice (Fig. 8b). Concomitantly, the CTCF-mediated proper chromatin conformation of the *Igf2-H19* domain forms the tissue-dependent insulation activity of maternal *H19*-ICR. However, the precise molecular mechanisms of OCT4/CTCF-mediated chromatin conformation during early development is still unknown. To understand this, further chromatin interaction analyses during pre- and post-implantation development must be performed. Findings from these further analyses will contribute to the improved understanding of not only the molecular pathogenesis of BWS but also other imprinting disorders, such as Kagami-Ogata syndrome, caused by gain of methylation at the ICR[45].

## Methods
### Mice
A mixture of 50 ng/μL sgRNA targeting SOBS (sgSO), 50 ng/μL Alt-R S.p. Cas9 nuclease V3 (1081058, IDT, Coralville, IA, USA), and 5 ng/μL ssODN (IDT) was injected into C57BL/6JJcl (B6, MGI:3055581, CLEA Japan, Tokyo, Japan) zygotes to generate *H19*-ICR[mSO] mice. After overnight culture, the embryos that developed to the two-cell stage were transferred into the oviducts of pseudopregnant Jcl:ICR female mice (MGI:5652875, CLEA

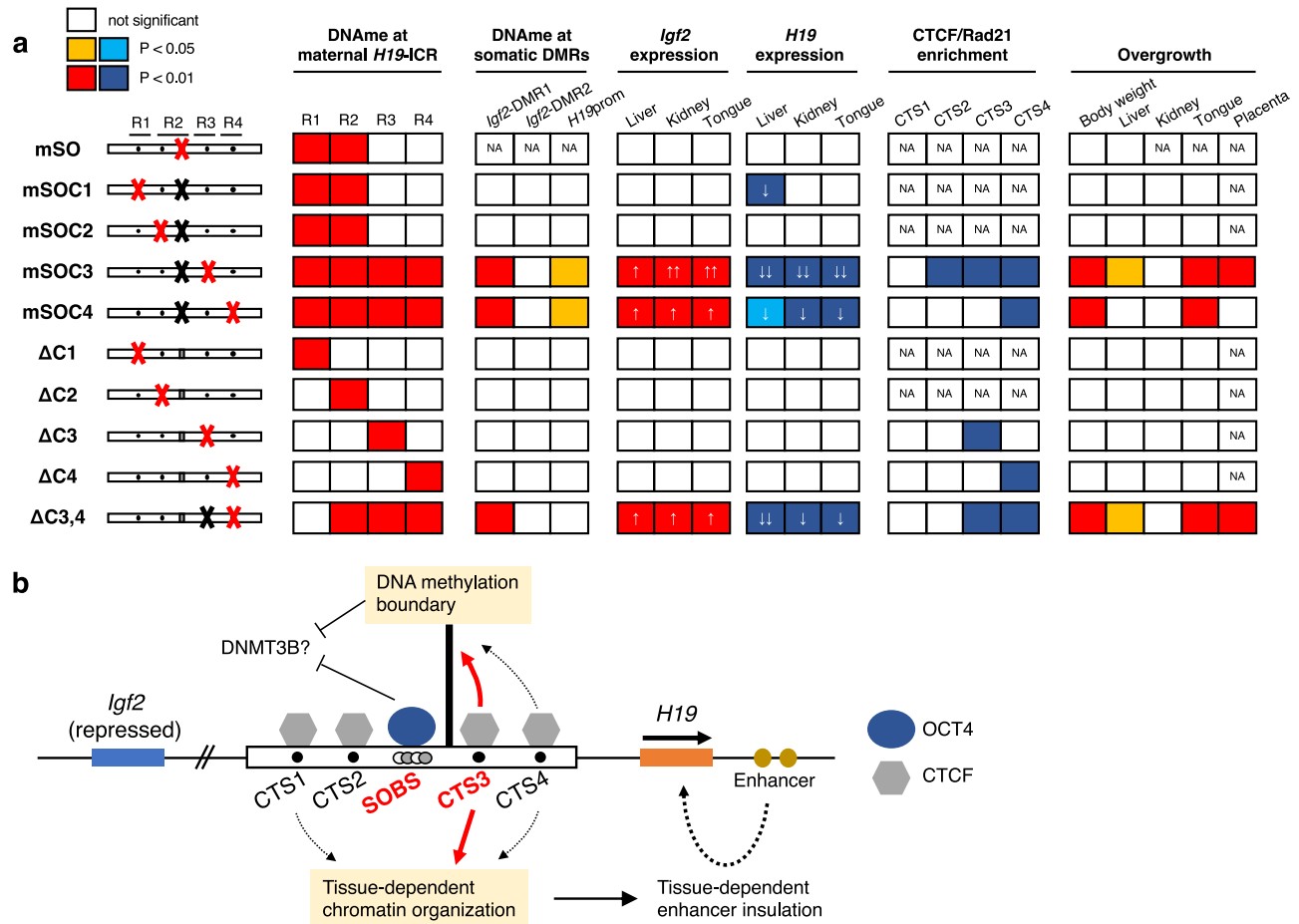

**Fig. 8 | Summary of this study. a** Summary of the results from mutant mice. Mutated sites in *H19*-ICR that each mutant mouse harbors are shown on the left side. Panels with significant differences are colored. Decreased levels of *H19* expression are indicated by one or two arrows. NA; not applicable. **b** Schematic model of mechanisms for the unmethylated state of maternal *H19*-ICR and imprinted expression of the *Igf2-H19* domain. Black line, boxes, circles, and arrows are similar to those described for Fig. 1A. CTCF binding on CTS3 forms DNA methylation boundary. OCT4 binding on SOBS and DNA methylation boundary coordinately protects DNA methylation during post-implantation development. In contrast, the binding of CTCF to CTS1/3/4 organizes liver-specific chromatin conformation together with the cohesin complex, resulting in liver-specific enhancer insulation activity. CTS3 is the main contributor to these functions.

Japan). For anesthesia, a freshly prepared mixture of ketamine HCl (Kyo-somirai Pharma, Tokyo, Japan) and xylazine HCl (Bayer Yakuhin, Osaka, Japan) diluted with sterile saline solution was used. The mice were anaes-thetized by intraperitoneal injection of 0.1 ml of the mixture per 10 g body weight equal to a dose of 100 mg/kg ketamine and 10 mg/kg xylazine. To generate *H19*-ICR$^{mSOC1-4}$ and *H19*-ICR$^{\Delta C1-4}$ mice, sperm from B6-background heterozygous *H19*-ICR$^{mSO}$ male mice were fertilized with B6 oocytes in vitro. The resulting zygotes were then microinjected with sgRNA targeting the respective CTS (sgCTS1-4) and Cas9 protein to obtain new-borns. As shown in Supplementary Fig. 13, mutant mice in which indel mutation was introduced at the mSO allele were defined as SOBS/CTS double mutants (*H19*-ICR$^{mSOC1-4}$). Contrastingly, those in which indel mutation was introduced at the SOBS-intact WT allele were defined as CTS single-mutant mice (*H19*-ICR$^{\Delta C1-4}$). To generate *H19*-ICR$^{\Delta C3,4}$ mice, sgCTS4/Cas9 was injected into zygotes fertilized with *H19*-ICR$^{+/\Delta C3}$ sperm and B6 oocytes. The target sequences of sgRNAs were searched using CRISPRdirect (https://crispr.dbcls.jp/)[46], and the sequence with the lowest number of off-target sites was used. All sgRNAs were prepared using the CUGA7 gRNA Transcription Kit (314-08691; Nippon Gene, Toyama, Japan) according to the manufacturer's instructions. Primer and ssODN sequences are listed in Supplementary Table 1. Founder mice were crossed with B6 mice to obtain F$_1$ pups. Nucleotide sequences of all mutant alleles were determined by genotyping of F$_1$ offspring. Heterozygous F$_1$ mutant mice were crossed with B6 or PWK/PhJ mice (PWK, MGI:2160654, the

Jackson Laboratory, Bar Harbor, ME, United States). Heterozygous F$_2$ mutants and their WT littermates were used for all experiments.

Genomic DNA was extracted from the fingertips or tail tips of mutant mice for genotyping. PCR was performed using BIOTAQ (BIO-21040; BioLine, London, UK) with the primers listed in Supplementary Table 2. The PCR products were treated with ExoSAP-IT (78201; Thermo Fisher Scientific, Waltham, MA, USA) and sequenced using a capillary sequencer (SeqStudio Genetic Analyzers; Applied Biosystems, Foster City, CA, USA). Nucleotide sequences of sgCTS1-4/Cas9-derived indel mutations in *H19*-ICR$^{mSOC1-4}$, *H19*-ICR$^{\Delta C1-4}$, and *H19*-ICR$^{\Delta C1-4}$ alleles are listed in Supple-mentary Fig. 14.

Mice were housed in a 12 h light/12 h dark cycle, and water and food were given *ad libitum*. Timed mating was conducted using 8–12-weeks-old male and female mice, with the morning of an observed copulation plug established as 0.5 days post coitum (dpc). Embryos/offspring were collected at 6.5 dpc, 10.5 dpc, 15.5 dpc, 16.5 dpc, and 1–21 days postpartum (dpp). The sex of the embryos/offspring was not recorded. Neonates at 1 dpp were euthanized by decapitation. For phenotypic analysis, liver, kidneys and tongue were removed and weighed. Tissues were then snap-frozen in liquid nitrogen for expression analyses. All dams were euthanized by cervical dislocation to obtain mutant embryos. Placenta and whole body were recovered from conceptuses of 15.5 and 16.5 dpc for phenotypic analysis. Embryos at 6.5 dpc were carefully isolated from decidua using a tungsten needle for DNA methylation analysis. Embryos at 10.5 dpc were

homogenized by BioMasher (49118-52; Nippi, Tokyo, Japan) in phosphate buffer saline containing proteinase inhibitor cocktail (11836170001; Sigma-Aldrich) for chromatin immunoprecipitation. We have complied with all relevant ethical regulations for animal use. All animal protocols were approved by the Animal Care and Use Committee of Saga University, Saga, Japan. All experiments were conducted in accordance with the Regulations on Animal Experimentation of Saga University.

**Expression analysis of imprinted genes.** Total RNA was isolated using ISOGEN II (311-07361; Nippon Gene, Tokyo, Japan) according to the manufacturer's instructions. Next, cDNA was synthesized using the PrimeScript RT reagent Kit with gDNA Eraser (RR047A; Takara Bio, Shiga, Japan) according to the manufacturer's instructions. Quantitative RT-PCR analysis of the imprinted genes was performed using the Taq-Man Fast Universal PCR Master Mix (4352042; Applied Biosystems). Relative quantification of imprinted gene expression was normalized to *Actb* expression levels and calculated using the ΔΔCt method. Following TaqMan probes were used: *Igf2* (Mm00439564_m1; FAM-MGB), *H19* (Mm01156721_g1; FAM-MGB), and *Actb* (Mm01205647_g1; VIC-MGB_PL).

For allelic expression analysis, the PCR products amplified from cDNA were treated with ExoSAP-IT and sequenced. Pyrosequencing analysis was performed using PyroMark Q24 (QIAGEN GmbH, Hilden, Germany) to evaluate quantitative allelic expression. SNP rs8246124 (*Igf2*) and rs255556403 (*H19*) were used to distinguish between C57BL/6 and PWK alleles. The primers used are listed in Supplementary Table 3.

**DNA methylation analysis.** DNA was extracted from the tail tips of embryos or neonates using phenol/chloroform extraction followed by ethanol precipitation. Two micrograms of DNA were subjected to sodium bisulfite conversion using an EZ DNA Methylation Kit (D5002; Zymo Research, Irvine, CA, USA). Genomic regions were subsequently amplified using EpiTaq HS (R110A; Takara Bio, Shiga, Japan) and cloned into the pGEM-T Easy vector (A1360; Promega, Madison, WI, USA) for bisulfite sequencing. The inserts were amplified through PCR using M13 primers and sequenced.

For the DNA methylation analysis, fully grown oocytes were collected from 4-week-old mice generated by crossing B6-background mutant females with PWK males. Mutant mice and their WT littermates at 44–46 h after injection with 5 IU pregnant mare serum gonadotropin (Serotropin; ASKA Pharmaceutical Company, Tokyo, Japan) were euthanized by cervical dislocation. Cumulus–oocyte complex was isolated from ovaries in M2 medium (M7167; Sigma-Aldrich, St. Louis, MO, USA) containing 100 μM 3-Isobutyl-1-methylxanthine (I5879; Sigma-Aldrich). Cumulus cells were completely removed through pipetting. For DNA methylation analysis of blastocysts, embryos at 3.5 dpc were produced via in vitro fertilization using oocytes from mutant mice and PWK sperm, as previously described[47]. Genomic DNA was extracted from oocytes and blastocysts were extracted as previously described[48]. Next, DNA samples (70–80 oocytes and 7–10 blastocysts per sample) were subjected to bisulfite conversion through 2 μg carrier RNA (QIAGEN). Nested PCR was performed using EpiTaq HS with the primers listed in Supplementary Table 4. The PCR products were cloned and sequenced as described above.

The methylation levels were analyzed using QUMA[49]. Multiple clones were removed by excluding identical bisulfite sequences (included in QUMA).

**Chromatin immunoprecipitation.** ChIP was performed as previously described[48]. Briefly, crosslinked chromatin was sheared using a sonicator (Covaris M220, Woburn, MA, USA). Chromatin was immunoprecipitated using the following antibodies: rabbit monoclonal anti-CTCF (#3418; Cell Signaling Technology, Danvers, MA, USA) and rabbit polyclonal anti-Rad21 (ab992; Abcam, Cambridge, UK). Finally, qPCR was performed using the THUNDERBIRD qPCR Mix (QPS-101; TOYOBO). The primers are listed in Supplementary Table 5.

**Statistics and reproducibility.** At least two biological replicates were used for all experiments. Statistical analyses were performed using Prism software (version 10.0.2; GraphPad Software, San Diego, CA, USA). Statistical analyses were performed using an unpaired two-tailed *t*-test, one-way ANOVA, or Fisher's exact test. The Mann-Whitney *U*-test (included in QUMA) was used for DNA methylation analysis. $P < 0.05$ was considered statistically significant. The sample numbers were described in each figure legend.

### Reporting summary
Further information on research design is available in the Nature Portfolio Reporting Summary linked to this article.

## Data availability
The data that support this study are available from the corresponding author upon reasonable request. Supplementary data (Source data) are provided with this paper.

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

## Acknowledgements

We thank the Analytical Research Center for Experimental Sciences, Saga University for experimental support. We also thank Editage (www.editage.jp) for English language editing. This study was supported in part by the Japan Society for the Promotion of Science KAKENHI grant (grant numbers JP24K02422, H.S.; JP21K19451, H.S.; JP20H03643, H.S.; JP23K07251, K.H.; JP23K05592, S.H.; and JP22KK0111, S.H.) and Kawano Masanori Memorial Public Interest Incorporated Foundation for Promotion of Pediatrics (S.H.); the Japan Agency for Medical Research and Development (grant numbers JP22ek0109587, H.S.; JP23ek0109674, H.S.; and JP23ek019672, H.S.); and the Ministry of Health, Labour and Welfare Program (grant number 23FC1052, H.S.).

## Author contributions

S.H. and H.S. designed the experiments. F.M. and S.K. performed microinjections. S.H. and H.Y. performed mouse crossing and genotyping analyses. S.H. performed DNA methylation, expression, and phenotypic analyses. S.H. and H.S. prepared the manuscript. M.K.-I., K.H., and H.S. reviewed and edited the manuscript.

## Competing interests

The authors declare no competing interests.
