## [Transparent Peer Review file · Communications Biology]

Identification of responsible sequences which mutations cause maternal H19-ICR hypermethylation with Beckwith–Wiedemann syndrome-like overgrowth

Corresponding Author: Professor Hidenobu Soejima

Version 0:

Reviewer comments:

Reviewer #1

(Remarks to the Author)
review – Hara *et al.*, Nature Communications

The manuscript by Hara *et al.* describes the identification of the sequence elements required for maintaining a differentially methylated state within the *H19* imprinting control region, and concomitantly, imprinted expression across this imprinting cluster in mice. The rationale behind this investigation is to better understand the elements that lead to gain of methylation (GOM) at the *H19*-ICR as GOM at this locus in humans results in Beckwith-Wiedemann syndrome. The experiments detailed in this manuscript represent a thorough and careful analysis of sequence elements, both individually and in combination, in order to parse the role of different sequences. Hara and colleagues demonstrate that while mutation of either the SOX2 and OCT4 binding sites (SOBS) or individual CTCF binding sites (CTSs) leads to hypermethylation of adjacent CpG dinucleotides, alone neither mutation impacts the expression of *Igf2* or *H19* nor do they result in phenotypic changes associated with BWS (weight gain). In contrast, simultaneous mutation of SOBS and CTCF binding site (CTS) 3 or 4 leads to significant hypermethylation across the ICR as well as at *Igf2*-DMR1 and the *H19* promoter. These broader epigenetic changes dramatically affect the expression of the imprinted genes in this cluster: *Igf2* expression increases due to activation of the previously silent paternal allele and *H19* expression decreases. Importantly, the altered epigenetic state and expression patterns have a phenotypic consequence: body weight is altered, result in BWS-like phenotypes. Hara and colleagues further investigated the developmental timing of the aberrant methylation and found that it occurs during post-implantation development, suggesting that the unmethylated state of the maternal allele, which is inherited from the oocyte, is not maintained when these sequences are mutated. While the effects of C3 and C4 mutations are strongest in combination with SOBS mutations, simultaneous mutation of C3 and C4 without mutation of SOBS also results in GOM and alteration of *Igf2* and *H19* gene expression levels, albeit to a lesser degree. Interestingly, the overgrowth phenotype is not as pronounced in this condition, suggesting synergy between SOBS and CTS3/4. Finally, the authors show evidence that CTCF and RAD21 binding to the maternal *H19*-ICR is disrupted when both SOBS and CTS3 are mutated, consistent with the GOM and providing a mechanistic explanation for the alteration in *Igf2* and *H19* expression patterns.

This manuscript presents solid evidence that the authors have identified sequences important for protecting the maternal *H19*-ICR from methylation during post-implantation development and that they have identified CTS3 as the DNA methylation boundary within this locus. The investigations are thorough, the data is clearly presented, and the interpretations are logical. Hara and colleagues further discuss tissue-specific effects that they identified that may explain why growth/overgrowth phenotypes may vary in their study as compared to previously published work. This work is valuable because it defines the functional relevance of specific sequences in an ICR; this type of thorough investigation has not been performed for the vast majority of the ICRs that have been identified. Of particular relevance is the fact that the authors have connected their functional dissection of the ICR sequence with chromatin conformation data supporting a TAD structure mechanism to explain why imprinted regulation of *Igf2* and *H19* expression is disrupted in when maternal ICR mutations are present. The findings detailed in this manuscript may lead to a better understanding of how GOM and BWS could result in humans.

Feedback:

1. In several places (Title, Abstract line 30, Introduction line 96, Results line 272, and Discussion lines 343 & 360), the authors state that SOBS and CTS3 are responsible for GOM at the *H19*-ICR, when their data illustrate that **mutation** of these sequences results in GOM. These statements should be rephrased to more clearly indicate that these sequences are important for preventing GOM at the *H19*-ICR and/or that mutation of these sequences leads to

GOM.

2. I very much appreciate that the authors analyzed two biological replicates and presented the biological replicate data so clearly in their supplemental figures. However, in some cases it feels like the exclusion of the biological replicate data from the in-text figures is a bit misleading as the replicates may also show significant differences from each other. This is most evident with the data shown in Supplementary Figure 4, where the methylation data looks dissimilar between the biological replicates ($\Delta C1/+ R1$ & *Igf2*-DMR1, $\Delta C2/+ R2$) and there may not be significance between one of the biological replicates and the WT. While I am comfortable with the overall interpretations and conclusions, the paper would benefit from some additional statistical analyses. For example, methylation levels of the two biological replicate samples could be compared to each other to illustrate whether they show any significant differences. If they do, each mutant biological replicate could be compared individually to the wild-type sample (in addition to the combined data comparison, which is already shown).

3. While this manuscript is generally very well written, the authors don't provide a very satisfying explanation for the variation in the growth/overgrowth phenotypes that they observed; this could be expanded upon and potentially connected to the methylation differences the authors observed in their biological replicate samples.

Reviewer #2

(Remarks to the Author)

Comments for the Authors

In this manuscript, Hara et al., investigated the DNA sequences necessary for maintaining hypomethylation of the maternal allele in the *Igf2*-H19 imprinted domain's ICR (H19-ICR), which is the causative region of BWS, using in vivo analysis in mice. They mutated one or two of the four CTCF-binding sites (C1, C2, C3, C4) and the SOX-OCT-binding site (SOBS) within the mouse endogenous H19-ICR, and examined the DNA methylation of the ICR, the imprinted expression of H19 and *Igf2*, and the presence of BWS-like phenotypes. They concluded that both SOBS and C3 are important for maintaining hypomethylation of the maternal H19-ICR and for normal fetal growth (suppression of overgrowth). The experimental data are reliable and sufficient, and I believe it is suitable for publication in Communications Biology.

Minor Comments

Regarding H19-ICR, as mentioned in the Introduction, various genetically modified mice have been created and analyzed. A discussion comparing the current experimental results with past data would help readers organize their thoughts. For example, the mice created by Ideraabdullah et al. (Ref# 23) lack the region containing OSBS and C3, so it is not surprising that they exhibit gene expression and phenotypes similar to mSOC3 of this manuscript. However, the neonatal weight gain observed in this manuscript is not as pronounced in the model mice created by Ideraabdullah et al. (Ref# 23). Can the authors discuss this point?

In mSOC3, the expression level of *Igf2* is increased, including in the kidney, but kidney hypertrophy does not occur. Can the authors discuss possible reasons for this?

In the mutant mice of Ref# 22 and Ref# 23, growth retardation similar to Silver-Russell syndrome (SRS) is observed with paternal transmission of the mutate alleles. According to ref# 21, SOX2 and OCT4 also bind to the highly methylated paternal SOBS. In this manuscript, Hara et al. only describe the maternal transmission of the created mutations, but it would be worthwhile to provide data or mention the paternal transmission of the mutations.

I feel that the concept of the DNA methylation boundary is insufficiently explained. It was thought that the high methylation between SOBS and C3 in $\Delta C3$ would disrupt the boundary function and hypermethylation would spread into R2 region, but the actual data show that hypermethylation of R2 does not occur in $\Delta C3$. I believe an explanation for this is necessary.

In Line 361, it is stated that the binding of OCT4 to SOBS contributes to the maintenance of hypomethylation. However, according to the ChIP-Atlas web site (https://chip-atlas.org/peak_browser), various transcription factors and epigenetic regulators such as NANOG, KMT2C, and OTX2 also bind around SOBS in mouse ES cells, in addition to SOX2 and OCT4. Considering that SOX2 and OCT4 bind regardless of methylation, could it be possible that the binding of DNA methylation-sensitive factors other than SOX2 and OCT4 is involved in maintaining hypomethylation of maternal H19-ICR?

Reviewer #3

(Remarks to the Author)

Hara et al. clarify which regions are critical for the protection of the mouse maternal H19-ICR from gain of methylation (GOM). Mutation of the SOX-OCT binding site (SOBS) alone does not gain methylation beyond the CTCF binding site3 (CTS3). On the other hand, mutations in CTS1-4 alone have little effect on DNA methylation, revealing that mutations in both SOBS and CTS3 result in a complete GOM of H19-ICR. The authors also show that GOM is increased in double mutants of CTS3 and CTS4. In order to demonstrate these phenomena, the authors have conducted very time-consuming experiments by creating mutant mice of SOBS and CTS1-4, and double mutants of CTS3 and CTS4. As a result, the authors present novel and detailed findings on the GOM of the H19-ICR. Although there is almost nothing to correct, the reviewer has a few relatively minor comments, explained below.

1) For SOBS, mutations are introduced by knock-in using donor ssODN, whereas for CTS1-4, mutations are introduced by NHEJ. The knock-in method, which can reproduce the same mutation each time, is considered more suitable in the present

study, especially when comparing single and multiple mutations. Of course, NHEJ-mediated mutations are acceptable, but why did the authors change to the NHEJ-mediated mutagenesis for CTS1-4?

2) L279: A ChIP analysis of RAD21 was performed, but there is little explanation about RAD21. It is necessary to explain in more detail the role of RAD21.

3) Fig.6 shows an increase in GOM for double-mutant compared to $\Delta C3$ and $\Delta C4$ alone. Although the data for $\Delta C3$ and $\Delta C4$ alone are shown in sup Figs, it would be better to show the data for $\Delta C3$ and $\Delta C4$ alone in Fig. 6 to enable comparison of the differences with the double-mutant.

4) L432-L433 : "Genomic DNA was extracted from oocytes and blastocysts as previously described" may be correct.

Version 1:

Reviewer comments:

Reviewer #1

(Remarks to the Author)

Review of the revised manuscript by Hara et al. has been conducted and all of this reviewer's concerns have been addressed in a satisfactory way. The additional statistical analyses and discussion have improved the manuscript.

Reviewer #2

(Remarks to the Author)

In this manuscript, Hara et al. investigated role of DNA sequences at the H19-ICR including SOX-OCT binding sites (SOBS) and the four CTCF-binding sites (C1, C2, C3, C4). They mutated the sequences one by one or two of them with multiple combinations in living mice. They showed that simultaneous mutations at the SOBS and C3 on the maternal H19-ICR disrupt hypomethylation state of the maternal H19-ICR and imprinted gene expressions of Igf2 and H19, leading BWS-like overgrowth phenotype. The experimental data are reliable and sufficient, and the responses to reviewer's comments are reasonable, therefore I believe it is suitable for publication in Communications Biology.

Reviewer #3

(Remarks to the Author)

I am satisfied with the revisions that have been made by authors.

Response to Reviewers

We are very grateful to the Reviewers for their careful evaluation and constructive suggestions of our manuscript. We have revised and improved our manuscript as suggested. Included below are our point-by-point responses to the comments and suggestions provided by the Reviewers.

Reviewer #1 (Remarks to the Author):

1. In several places (Title, Abstract line 30, Introduction line 96, Results line 272, and Discussion lines 343 & 360), the authors state that SOBS and CTS3 are responsible for GOM at the *H19*-ICR, when their data illustrate that **mutation** of these sequences results in GOM. These statements should be rephrased to more clearly indicate that these sequences are important for preventing GOM at the *H19*-ICR and/or that mutation of these sequences leads to GOM.

Response: Thank you for your careful evaluation of our manuscript. We have checked the entire manuscript and rephrased these statements to reflect this in the revised manuscript (Title, Abstract line 30, Introduction line 93, Results line 282, and Discussion lines 379 and 393).

2. I very much appreciate that the authors analyzed two biological replicates and presented the biological replicate data so clearly in their supplemental figures. However, in some cases it feels like the exclusion of the biological replicate data from the in-text figures is a bit misleading as the replicates may also show significant differences from each other. This is most evident with the data shown in Supplementary Figure 4, where the methylation data looks dissimilar between the biological replicates ($\Delta C1/+$ R1 & *Igf2*-DMR1, $\Delta C2/+$ R2) and there may not be significance between one of the biological replicates and the WT. While I am comfortable with the overall interpretations and conclusions, the paper would benefit from some additional statistical analyses. For example, methylation levels of the two biological replicate samples could be compared to each other to illustrate whether they show any significant differences. If they do, each mutant biological replicate could be compared individually to the wild-type sample (in addition to the combined data comparison, which is already shown).

Response: Thank you for this important suggestion. We conducted statistical analysis between the biological replicates using QUMA. In addition, each mutant biological replicate was compared individually to the WT. We found that DNA methylation levels in the

methylation data, as mentioned by Reviewer#1 (R1 and *Igf2*-DMR1 in *H19*-ICR^{ΔC1/+}, R2 in *H19*-ICR^{ΔC2/+}), were significantly different between biological replicates, but that for all other locations were not significantly different. This suggests that a mutation at a single CTS causes the variation of DNA methylation around mutated CTS. The results of statistical analyses have been added in Supplementary Figures 3b, 3c, 4b, and 4c of the revised manuscript. To more accurately reflect this, we have modified the Discussion from lines 371–378 in the revised manuscript.

3. While this manuscript is generally very well written, the authors don't provide a very satisfying explanation for the variation in the growth/overgrowth phenotypes that they observed; this could be expanded upon and potentially connected to the methylation differences the authors observed in their biological replicate samples.

Response: Our results indicate that the frequency of overgrowth was different among various mutant mice, and the severity of overgrowth was different in individuals harboring the same genotype. The variety of phenotypes observed in patients with BWS is attributable to mosaicism between normal and *H19*-ICR GOM cells (Ref #8). In addition, DNA methylation levels at maternal *H19*-ICR and *Igf2*-DMR1 were varied in some mutant mice (Supplementary Figure 3b–c and 4b–c), suggesting that mosaicism of normal and *H19*-ICR GOM cells was observed in mutant mice as well as in humans. Thus, stochastic epigenetic mechanisms may be responsible for the pathogenesis of overgrowth in mutant mice. We have added this information to the Discussion section on lines 371–378 in the revised manuscript.

Reviewer #2 (Remarks to the Author):

Minor Comments

Regarding *H19*-ICR, as mentioned in the Introduction, various genetically modified mice have been created and analyzed. A discussion comparing the current experimental results with past data would help readers organize their thoughts. For example, the mice created by Ideraabdullah et al. (Ref# 23) lack the region containing OSBS and C3, so it is not surprising that they exhibit gene expression and phenotypes similar to mSOC3 of this manuscript. However, the neonatal weight gain observed in this manuscript is not as pronounced in the model mice created by Ideraabdullah et al. (Ref# 23). Can the authors discuss this point?

Response: Mice generated by Ideraabdullah et al. ($H19^{\Delta C2,3/+}$) harboring a deletion including SOBS, CTS2, and CTS3 showed unmethylated status of the remaining CTSs on the maternal allele (CTS1 and CTS4) and smaller effects than $H19-ICR^{mSOC3/+}$. Since CTS spacing was reported to alter CTCF-dependent insulation activity, the deletion in $H19^{\Delta C2,3/+}$ might alter the spacing between CTS1 and CTS4, and rescue maternal-specific TAD formation in $Igf2-H19$. We have added this information to the Discussion section on lines 324–329 of the revised manuscript.

In $mSOC3$, the expression level of $Igf2$ is increased, including in the kidney, but kidney hypertrophy does not occur. Can the authors discuss possible reasons for this?

Response: As for the kidney in these mutants, overgrowth was not observed despite both the increased $Igf2$ expression and the decreased $H19$ expression. Increased kidney weight has been observed in chimeric mice with transgenic ES cell lines that highly express $Igf2$ (Ref #33), suggesting that the dosage of $Igf2$ in the kidney of our mutant mice was not sufficient to induce overgrowth of the kidney. We have included a description of this in the Discussion section on line 350–352 of the revised manuscript.

In the mutant mice of Ref# 22 and Ref# 23, growth retardation similar to Silver-Russell syndrome (SRS) is observed with paternal transmission of the mutant alleles. According to ref# 21, SOX2 and OCT4 also bind to the highly methylated paternal SOBS. In this manuscript, Hara et al. only describe the maternal transmission of the created mutations, but it would be worthwhile to provide data or mention the paternal transmission of the mutations.

Response: We generated mice with paternally transmitted $H19-ICR^{mSOC3}$ allele ($H19-ICR^{+/mSOC3}$) by crossing $H19-ICR^{mSOC3}$ male mice with PWK females. Our results showed that the DNA methylation at paternal $H19-ICR$, the expression of imprinted genes, and the phenotypes were indistinguishable from their WT littermates. These results indicate that SOBS and CTS3 mutations do not cause imprinting defects at paternal $H19-ICR$. We have added a paragraph in the Results section (lines 225–235 in the revised manuscript). Also, we have added these data in Supplementary Figure 7. As a result of this modification, Supplementary Figures 7–13 in the original manuscript are now renamed Supplementary Figures 8–14.

I feel that the concept of the DNA methylation boundary is insufficiently explained. It was thought that the high methylation between SOBS and C3 in $\Delta C3$ would disrupt the boundary function and hypermethylation would spread into R2 region, but the actual data

show that hypermethylation of R2 does not occur in $\Delta C3$. I believe an explanation for this is necessary.

Response: Thank you for your insightful feedback. Mutation of CTS3 alone did not extend DNA methylation toward the R2 region. Moreover, *H19-ICR $\Delta C3/+$* neonates showed partial maternal *H19-ICR* GOM and did not exhibit any imprinting defects. These suggest that the maternal-specific TAD structure, which is rescued by CTCF binding on remaining CTSs, limits the expansion of *H19-ICR* GOM in *H19-ICR $\Delta C3/+$* . In addition, SOBS was completely methylated at the R2 region in *H19-ICR $\Delta C3,4/+$* neonates, but with little spreading toward CTS1 and CTS2. This implies that another methylation boundary might exist between SOBS and CTS2. We have added this point to the Discussion section in lines 330–335 in the revised manuscript.

In Line 361, it is stated that the binding of OCT4 to SOBS contributes to the maintenance of hypomethylation. However, according to the ChIP-Atlas web site (https://chip-atlas.org/peak_browser), various transcription factors and epigenetic regulators such as NANOG, KMT2C, and OTX2 also bind around SOBS in mouse ES cells, in addition to SOX2 and OCT4. Considering that SOX2 and OCT4 bind regardless of methylation, could it be possible that the binding of DNA methylation-sensitive factors other than SOX2 and OCT4 is involved in maintaining hypomethylation of maternal *H19-ICR*?

Response: Thank you for your insightful suggestion. We agree with the reviewer's comment. In ES cells, it is not only OCT4 and SOX2 but also other transcription factors and epigenetic regulators are also enriched around SOBS in *H19-ICR*. However, it is largely unknown as to whether the DNA binding activity of these proteins is dependent on DNA methylation, and further studies are required to clarify this. We have detailed this in the Discussion section on line 365–370 in the revised manuscript.

Reviewer #3 (Remarks to the Author):

1) For SOBS, mutations are introduced by knock-in using donor ssODN, whereas for CTS1-4, mutations are introduced by NHEJ. The knock-in method, which can reproduce the same mutation each time, is considered more suitable in the present study, especially when comparing single and multiple mutations. Of course, NHEJ-mediated mutations are acceptable, but why did the authors change to the NHEJ-mediated mutagenesis for CTS1-4?

Response: As Reviewer#3 mentioned, induction of homology-directed repair (HDR)-mediated point mutations at each CTS will allow for more precise effects of CTS mutations. However, the efficiencies of genome editing using sgCTS1 and sgCTS2 were lower than those of sgCTS3 and sgCTS4, thus making it difficult to obtain double mutations with mSO. For this reason, we elected to introduce mutations by NHEJ in all mice.

2) L279: A ChIP analysis of RAD21 was performed, but there is little explanation about RAD21. It is necessary to explain in more detail the role of RAD21.

Response: RAD21 is known to be a subunit of the cohesin complex. Recent studies have reported that RAD21 facilitates cohesin loading to the genome and formation of the 3D chromatin loop. We have added molecular detail of RAD21 on lines 290–291 of the revised manuscript. According to this modification, we have also included reference #25 in the revised manuscript.

3) Fig.6 shows an increase in GOM for double-mutant compared to $\Delta C3$ and $\Delta C4$ alone. Although the data for $\Delta C3$ and $\Delta C4$ alone are shown in sup Figs, it would be better to show the data for $\Delta C3$ and $\Delta C4$ alone in Fig. 6 to enable comparison of the differences with the double-mutant.

Response: Thank you for this important suggestion. As Reviewer#3 states, it is very important to compare the $\Delta C3$ and $\Delta C4$ series with $\Delta C3,4$. However, the main focus of this experiment is to compare the results of maternal transmission of $\Delta C1-4$, in which CTS is mutated alone, with those of mSOC1-4, in which CTS is mutated in combination with SOBS. Moreover, $\Delta C3,4$, which simultaneously mutates CTS3/CTS4, was created with the main objective of comparing it with mSOC3 and mSOC4, which showed similar phenotypes. Therefore, we have determined that the current configuration of Figure 6 should remain as it is.

4) L432-L433: "Genomic DNA was extracted from oocytes and blastocysts as previously described" may be correct.

Response: We have corrected this on line 465 of the revised manuscript.

Other points revised in this version:

1) In the course of revising this manuscript, we noticed an error where the DNA methylation of mSOC2#1 and $\Delta C3$ #1 were used in the same image in the original manuscript; therefore, we replaced the result of mSOC2#1 with the correct image in

Figure 3 and Supplementary Figure 4 of the revised manuscript. We note that this change does not alter the results of the statistical analysis or the conclusions of the paper.

- 2) We have checked the figure lettering and changed it to lowercase in the revised manuscript.